Arousal modulates the motor interference effect stimulated by pictures of threatening animals

Cao Gai
Liu Peng liupeng@nwu.edu.cn
School of Public Administration/ School of Emergency Management, Northwest University , Xi’an , Shaanxi Province , China
Gollo Leonardo
Electronic publication date: 2021 Feb 11
Publication date: 2021
Volume: 9
Electronic Location ID: e10876
Received 2020 Aug 27; Accepted 2021 Jan 11
Copyright: ©2021 Cao and Liu
Copyright year: 2021
Copyright holder: Cao and Liu
License: This is an open access article distributed under the terms of the Creative Commons Attribution License, which permits unrestricted use, distribution, reproduction and adaptation in any medium and for any purpose provided that it is properly attributed. For attribution, the original author(s), title, publication source (PeerJ) and either DOI or URL of the article must be cited.
License URL: https://creativecommons.org/licenses/by/4.0/

Keywords: Motor interference effect, Motor priming paradigm, Threatening animals, Arousal, Valence

Funding: The Scientific Research Program funded by the Shaanxi Provincial Education Department 19JK0827 This work was supported by the Scientific Research Program funded by the Shaanxi Provincial Education Department (No. 19JK0827). The funders had no role in study design, data collection and analysis, decision to publish, or preparation of the manuscript.

==============================
Previous research related to the motor interference effect from dangerous objects indicated that delayed responses to dangerous objects were associated with more positive parietal P3 amplitudes, suggesting that great attentional resources were allocated to evaluate the level of danger (i.e., negative valence). However, arousal covaried with valence in this research. Together with previous studies in which the P3 amplitude was found to be increased along with a higher arousal level in the parietal lobe, we raised the issue that more positive parietal P3 amplitudes might also be affected by a high arousal level. To clarify whether valence or arousal impacted the motor interference effect, this study used a motor priming paradigm mixed with a Go/NoGo task and manipulated the valence (negative, neutral and positive) and arousal (medium and high) of target stimuli. Analysis of the behavioral results identified a significant motor interference effect (longer reaction times (RTs) in the negative valence condition than in the neutral valence condition) at the medium arousal level and an increased effect size (increment of RT difference) at the high arousal level. The results indicated that negative valence stimuli may interfere with the prime elicited motor preparation more strongly at the high arousal level than at the medium arousal level. The ERP results identified larger centroparietal P3 amplitudes for the negative valence condition than for the neutral valence condition at a high arousal level. However, the inverse result, i.e., lower centroparietal P3 amplitudes for the negative valence condition than for the neutral valence condition, was observed at a medium arousal level. The ERP results further indicated that the effect size of the behavioral motor interference effect increased because subjects are more sensitive to the negative valence stimuli at the high arousal level than at the medium arousal level. Furthermore, the motor interference effect is related to the negative valence rather than emotionality of the target stimuli because different result patterns emerged between the positive and negative valence conditions. Detailed processes underlying the interaction between valence and arousal effects are discussed.

Introduction

Increasing production safety accidents occur in factories when workers interact with more types of machines. For example, during the process of operating a machine, dicing or saw blades in the machine may cut off a worker’s finger if his or her prepared motor actions are not inhibited in time. According to a survey, human factors cause approximately 88% of work-related accidents (Huang et al., 2012). Thus, it is necessary to explore how we process our prepared motor actions to avoid touching dangerous elements in a machine, which might provide a reference for safety management that reduces the occurrence of work-related accidents from a cognitive perspective.

There has been evidence that dangerous objects may delay an individual’s response speed; this phenomenon is known as the motor interference effect from dangerous objects (Anelli, Borghi & Nicoletti, 2012). Using a motor priming paradigm, previous research has investigated mechanisms of the motor interference effect from behavioral and event-related potential (ERP) perspectives. Anelli, Borghi & Nicoletti (2012) adopted a motor priming paradigm using different means of grasping right hands as primes (a grasping or static human hand or a grasping robotic hand) and dangerous or safe objects as targets. Participants were asked to respond to artifacts or natural attributes of the targets. The results showed delayed responses for dangerous targets compared with those for safe targets, suggesting that the threat imposed by an object may conflict with an individual’s prepared motor actions and thus cause slower response times (RTs). To further investigate the neuro-origin of the motor interference effect from dangerous objects, Liu et al. (2017) adopted a motor priming paradigm mixed with a Go/NoGo task. Left or right grasping hands were used as prime stimuli, and dangerous (a rectangular saw blade and a round saw blade) or safe (a ruler and a disc) objects were used as target stimuli. A green or red dot was superimposed on the center of the targets as a Go or NoGo signal, respectively. The participants were asked to prepare the ipsilateral hand responses that corresponded to the handedness of the prime. They were asked to execute the prepared responses until the green dot (Go signal) emerged and to inhibit their responses while they observed the red dot (NoGo signal). The behavioral results replicated a classical motor interference effect from dangerous objects, as found by Anelli, Borghi & Nicoletti (2012). Moreover, the ERP results revealed a more positive parietal P3 amplitude in a dangerous condition than in a safe condition. As the parietal P3 component represents attentional resource allocation (Isreal et al., 1980), Liu et al. (2017) concluded that the motor interference effect originated from danger evaluation because many attentional resources were recruited to evaluate the dangerousness of the target to prevent the touching of dangerous stimuli.

The motor interference effect from dangerous objects can be explained by the notion that a dangerous target activates an aversive motivational system that may elicit an avoidance response. The activation of the aversive motivational system is deemed to be related to the valence attribute of stimuli (Lang, Bradley & Cuthbert, 1997). Valence reflects subjective appraisal of stimulus pleasantness (positive versus negative); specifically, a more negative valence stimulus may elicit a deeper activation of the aversive motivational system (Lang, Bradley & Cuthbert, 1990; Lang, Davis & Öhman, 2000). However, another important attribute (i.e., arousal, reflecting the level of activation sensitivity from calmness to excitation) of threatening stimuli was neglected by Liu et al. (2017) according to the dimensional models of emotions (Russell, 1980), resulting in the issue that arousal level covaried with valence factor. The dangerous targets (rectangular or circular sawblades) might not only activate an aversive motivational system (elicited by negative valence of the dangerous targets) but also elicit a high-level excitation state (elicited by a high arousal level of the dangerous targets) because attentional resources are automatically recruited to process the dangerous targets (Blanchette, 2006; Fox et al., 2000; Öhman, Flykt & Esteves, 2001; Tipples et al., 2002). However, the safe targets (i.e., rule or disc) might not activate the aversive motivational system and only elicit a medium-level excitation state (elicited by a medium arousal level of the safe targets). Furthermore, existing evidence indicates that amplitudes of parietal P3 components increase along with a higher level of arousal (Cuthbert et al., 2000; Keil et al., 2002; Polich, 2007; Sabatinelli et al., 2007; Schupp et al., 2003). Therefore, the presence of more positive parietal P3 amplitudes in dangerous conditions compared with those in safe conditions, reported by Liu et al. (2017), could also be explained by the notion that the arousal level of dangerous targets was higher than that of safe targets. Additionally, an alternative explanation of the behavioral motor interference effect might still exist because positive valence stimuli were not involved in the study by Liu et al. (2017). Specifically, there has been evidence that, compared with process neutral valence stimuli, both negative and positive valence stimuli processing could increase attentional resources allocated to a task (Cuthbert et al., 2000; Keil et al., 2002). One might argue that the occurrence of the motor interference effect might be attributed to emotionality (either negative or positive valence) rather than negative valence of background stimuli because both negative and positive valence background stimuli could distract attention from the main Go/NoGo task, which might increase the RTs in dangerous conditions, as in the study by Liu et al. (2017).

To clarify whether manipulating the valence factor confounded the arousal levels in Liu et al. (2017), arousal levels should be matched when comparing results between negative and neutral valence conditions. Existing evidence indicates that arousal could modulate the valence effect on a cognitive task (Zsido et al., 2019; Zsido et al., 2020). For example, Zsido et al. (2020) utilized a novel paradigm in which participants were instructed to search for numbers in a matrix. The number matrix was presented as superimposed on an emotional picture. They manipulated factors of valence (negative, neutral and positive levels) and arousal (medium and high levels) of the emotional pictures to investigate which of the two factors had a greater impact on cognitive processing. The results indicated a significant interaction between arousal and valence factors. At the medium arousal level, performance was worse in the negative valence condition than in the neutral and positive valence conditions. However, at the high arousal level, performance of the negative valence condition was improved and became similar to the performance of the neutral and positive valence conditions. The results suggested that arousal could significantly modulate the valence effect on a cognitive task. Negative valence background stimuli could distract attention from the main cognitive task at the medium arousal level, which impaired performance in this condition. The decrement in performance could be compensated with increasing arousal levels (at the high arousal level) of the background stimuli because an increased arousal level could speed up the cognitive system, which in turn improved task performance in negative valence conditions to a level similar to that in neutral and positive valence conditions at the high arousal level.

By referring to the studies from Zsido’s lab, we systematically manipulated factors of valence and arousal to investigate whether these two factors impact the motor interference effect from dangerous objects. The study continued to use the experimental paradigm (a motor priming paradigm mixed with a Go/NoGo task) adopted by Liu et al. (2017) with the following exceptions: (1) The background targets were selected from the International Affective Picture System (IAPS) (Lang, Bradley & Cuthbert, 2005), in which levels of valence and arousal factors have been standardized and defined. Only touchable animal pictures were selected for the background target stimuli because touchable stimuli could interact with the motor preparations induced by a priming stimulus (a left or right grasping hand). Moreover, selecting animal pictures as the targets could also exclude a confounding factor that might be caused by different categories of targets. (2) Valence (negative, neutral and positive) and arousal (medium and high) levels of target stimuli were systematically manipulated according to Zsido et al. (2020). Positive valence was included to clarify whether the motor interference effect from dangerous objects occurs because negative valence implies delayed responses for background targets or whether it occurs because emotionality (either negative or positive) of target stimuli distracted attention from the Go/NoGo task, which in turn delays responses. Of note, the arousal levels were matched among negative, neutral and positive valence in the animal category of target stimuli, and the mean scores of the high arousal level (mean = 5.68) are lower than those of in the study by Zsido et al. (2020) (mean = 7.57). The effectiveness of the manipulation is discussed in the General Discussion. (3) Emotional pictures were used as backgrounds with a Go/NoGo signal (yellow capital letters “M” or “W”) superimposed on the center. Participants were instructed to prepare the ipsilateral hand responses that corresponded to the handedness of the prime and to decide whether to execute the prepared responses according to the Go/NoGo signal. Responding to the superimposed targets imitated a situation in which executing a prepared response encountered an emergent dangerous stimulus because emotion is triggered by accidental stimuli in usual life settings (Delplanque et al., 2005; Yuan et al., 2007). The current design that did not require subjects to evaluate emotion overtly (the emotional stimuli are used as backgrounds) may have allowed emotional responses in the laboratory setting to more closely imitate life experiences.

The hypotheses were based on the work of Zsido et al. (2020), namely, that arousal could modulate the strength of the motor interference effect. Specifically, at the medium arousal level, the mean RTs in the negative valence condition should be longer than those in the neutral and positive valence conditions because negative valence background stimuli could distract attention from the Go/NoGo signals at the medium arousal level. In contrast, high arousal stimuli might divert attentional resources to the background targets, which may enhance perceptual representation of negative valence targets. Participants might suppress their prepared responses more strongly because many threatening details could be analyzed by the increased attentional resources in the high arousal and negative valence condition. Therefore, a larger effect size (mean RT in the negative valence condition minus those in the neutral and positive valence conditions) should emerge at the high arousal level compared with the medium arousal level. For ERPs, according to Liu et al. (2017), who suggested that the valence factor could significantly contribute to the P3 amplitudes, more positive P3 amplitudes should emerge in the negative valence condition than in the neutral and positive valence conditions at the medium arousal level. Moreover, the differences in the P3 amplitudes between the negative and neutral valence conditions and those between the negative and positive valence conditions should be larger at the high arousal level than the medium arousal level.

Method

Participants

To obtain robust results, 76 right-handed subjects were recruited to participate in the experiment. Note that the number of subjects was more than that of a prior simple size estimation (seven subjects) with a large effect size (f = 0.4) and 0.95 statistical power. Subjects No. 6 and No. 11 were excluded from the data analysis because of a reference channel error. The remaining 74 subjects (thirty males), ranging in age from 18 to 26 years (mean age = 21.91), were included in the data analysis. All participants had normal or corrected-to-normal visual acuity. They also reported an absence of neurological disorders. They provided written informed consent before participating the experiment and were compensated with RMB 50 while finishing the experiment. The experiment was approved by the Medical Ethics Committee at Northwest University.

Materials and apparatus

Left or right hand with a partial forearm pictures (subtending a visual angle 13° horizontally and 11° vertically) were used as primes, which aimed to activate a corresponding directional response readiness (Liu et al., 2017). To imitate a spatially matched grasping situation, primes were presented 2° to the left or right of the fixation point. Centrally presented targets were combined pictures with a Go/NoGo signal (yellow capital letters “M” or “W” subtending a visual angle 2° horizontally and 2° vertically) superimposed on the center of an emotional picture (the size of raw pictures was adjusted to a visual angle 12° horizontally and 9° vertically). Emotional pictures were selected from the IAPS (Lang, Bradley & Cuthbert, 2005) with the following criteria: (1) touchable stimuli were selected as targets to interact with the motor preparations induced by priming stimuli and (2) all of the selected stimuli belonged to one category (i.e., animal) to exclude stimulus categories as a confounding factor. Accordingly, we selected 18 animal pictures (Table 1) as backgrounds with systematically manipulated valence and arousal levels (Table 2). Two two-way analyses of variances (ANOVAs), as a function of valence and arousal, were separately performed for the valence and arousal scores. Analysis of the valence scores revealed a significant main effect of valence [F(2, 12) = 65.94, p < 0.001, η2p = 0.92], together with non-significant main effect of arousal [F(1, 12) = 0.001, p = 0.99, η2p = 0.001] and interaction [F(2, 12) = 0.04, p = 0.97, η2p = 0.006]. Moreover, analysis of the arousal scores revealed a significant main effect of arousal [F(1, 12) = 42.13, p < 0.001, η2p = 0.78], together with non-significant main effect of valence [F(2, 12) = 0.67, p = 0.53, η2p = 0.10] and interaction [F(2, 12) = 0.08, p = 0.92, η2p = 0.01]. The results suggested that both valence and arousal levels were systematically manipulated, with equivalent valence ratings between medium and high arousal levels and equivalent arousal ratings among negative, positive and neutral valence levels.

Table 1 IAPS codes for target pictures in each valence and arousal level.

	Negative valence	Neutral valence	Positive valence	
Medium arousal	1230 (Spider)	1121 (Lizard)	1463 (Kittens)	
	1270 (Roach)	1313 (Frog)	1540 (Cat)	
	1275 (Roaches)	1947 (Octopus)	1590 (Horse)	
High arousal	1019 (Snake)	1560 (Hawk)	1650 (Jaguar)	
	1205 (Spider)	1640 (Coyote)	1710 (Puppies)	
	1301 (Dog)	1726 (Tiger)	1720 (Lion)	

Table 2 Means and standard deviations of valence and arousal scores as a function of valence and arousal.

The raw scores are derived from Table 1 in Lang, Bradley & Cuthbert (2005).

	Negative valence		Neutral valence		Positive valence	
	valence scores	arousal scores		valence scores	arousal scores		valence scores	arousal scores	
Medium arousal	3.69 (0.40)	4.81 (0.04)		5.76 (0.10)	4.52 (0.27)		7.26 (0.17)	4.69 (0.13)	
High arousal	3.77 (0.16)	5.78 (0.01)		5.68 (0.78)	5.62 (0.56)		7.26 (0.94)	5.65 (0.50)	

Data were recorded as previously described in Liu et al. (2017). Specifically, stimulus presentation was driven by an E-Prime software (version 2.0, Psychology Software Tools, Inc. Pittsburgh, PA, USA) on a standard PC linked to a 17- inch CRT monitor (60-Hz refresh rate). Electroencephalogram (EEG) data were recorded by a NeuroScan system (NeuroScan, Inc.). A Neuroscan Synamp 2 amplifier with a 64 Ag/AgCl electrode cap mounted according to an extended international 10–20 system was used to continuously record EEG data.

Procedure

To reduce eye fatigue, all stimuli were presented on a black background. Each trial was initiated from a 300-ms fixation cross to alert participants to concentrate on the screen. Then, a 300-ms blank screen, a 200-ms left- or right-hand prime, another 50-ms blank screen, and a 1000-ms target were successively presented. Note that the target display was terminated if the response was executed within 1,000 ms. The intertrial interval was randomized within 1,400–1,600 ms.

The participants were seated in front of a laboratory table in a dimly lit chamber. A bracket fixed on the table held their chin to maintain central eye fixation and to fix a computer screen placed 60 cm in front of their eyes throughout the experiment. They were instructed to respond according to the Go/NoGo signal (“M” or “W”), which was superimposed on the emotional pictures. Specifically, they were instructed to prepare a left- or right-hand response that corresponded to the handedness of the prime and not to execute the response until a Go signal appeared. Half of the participants were asked to execute the prepared response as fast and accurately as possible whenever the letter “M” (Go) emerged and to withhold the prepared response when presented with the letter “W” (Nogo). The response rule was counterbalanced in the other half of the participants, with the letter “W” as a Go signal and the letter “M” as a NoGo signal. The participants were instructed to execute a left-hand response by pressing the “F” key using the index finger of their left hand and to execute a right-hand response by pressing the “J” key using the index finger of their right hand on an English keyboard. In contrast, when a NoGo signal appeared, the participants were instructed to withhold the prepared response.

The design manipulated valence (negative, neutral and positive), arousal (medium versus high) and Go/NoGo factor (Go versus NoGo). The formal task contained 1152 trials, which included 3 levels of valence ×2 levels of arousal ×2 levels of Go/NoGo ×96 repetitions. In each condition, left or right handedness of the prime was assigned in equal proportions. At the beginning, a 12-trial practice was performed. The formal task would begin unless the participant correctly answered over 11-trials in the practice phase. Each block contained 128 trials. The participants could take a rest between the blocks until they were ready for the next block.

EEG recording and processing

The EEG data were recorded with the signals bandpass-filtered at 0.05–100 Hz and referenced to the tip of the nose. To ensure signal quality, the impedance of the electrodes was maintained at less than 5 kΩ throughout the experiment. The sampling rate was 500 Hz. The recorded EEG data were preprocessed using the EEGLAB toolbox (Delorme & Makeig, 2004) according to the following steps: (1) The continuous EEG data were resampled at 250 Hz. (2) The resampled EEG data were high-pass filtered at 0.1 Hz and low-pass filtered at 30 Hz. (3) The EEG data were segmented and time-locked to the target onset. The duration of each epoch was 3,000 ms with a baseline of 1,000 ms before the target onset. (4) The epoched data were corrected using the mean amplitude of the baseline. (5) The behavioral data were merged into the epoched data, and the incorrect trials were deleted. (6) Bad channels were deleted. (7) The epoched data contaminated by eye blinks and eye movements were corrected using the independent component analysis (ICA) algorithm (Delorme & Makeig, 2004). (8) The deleted channels were interpolated using the EEGLAB toolbox. (9) The epochs were re-referenced to the mean of the bilateral mastoid electrodes, and (10) the epochs with large artifacts were detected by eye and manually deleted. Automatic artifact detection was then performed with deletion of the trials containing amplitudes less than −100 µV or more than 100 µV. Consequently, 2.5% of the epochs were rejected as contaminated during preprocessing across all subjects and conditions. The mean number of artifact-free trials obtained for each condition stabilized between 93 and 94, which ensured a valid trial amount for each condition.

Before the final averaging, the preprocessed data were resegmented and initiated from 300 ms before and 900 ms after the target onset. A flat time window (−300 to −200 ms before the target onset) was selected as a baseline to correct the new epochs. Then, the extracted average waveforms for each participant and condition were used to calculate the grand-average waveforms.

Statistical analysis

Behavioral data

Mean RTs and mean error rates for each condition were averaged separately for each participant. Note that only the RTs for correct responses in Go trials were involved in the RT analysis. Before the analysis, Kolmogorov–Smirnov tests of normality were performed on RTs and error rates for each condition. The results indicated that mean RTs were normally distributed for all conditions. Accordingly, the mean RTs were analyzed by a two-way repeated-measures ANOVA as a function of valence (negative, neutral and positive) and arousal (medium versus high). However, the Kolmogorov–Smirnov tests indicated that the mean error rates deviated from normality for all conditions (p s < 0.01). Accordingly, logarithmic mean correct rates (because 0 cannot be log-transformed, we used correct rates instead of error rates) were analyzed by a three-way repeated-measures ANOVA as a function of valence (negative, neutral and positive), arousal (medium versus high) and Go/NoGo factor (Go versus NoGo).

ERP data

To utilize more channel signals, the centroparietal scalp regions of interest (SROIs, the average of the C5, C3, C1, Cz, C2, C4, C6, CP5, CP3, CP1, CPz, CP2, CP4, CP6, P5, P3, P1, Pz, P2, P4 and P6 electrodes) were defined according to the topographic maps (more details are presented in Fig. 1) and a previous study in which a similar task (a Go/NoGo signal superimposed on an emotional picture) was adopted as the current study (Zhao et al., 2019). The dependent variables comprised centroparietal P3 amplitudes, which were calculated based on the mean amplitude between 300 and 500 ms for Go and that between 400 and 600 ms for NoGo trials. Note that only P3 amplitudes were used as an ERP index because Liu et al. (2017) identified that the neural processing of the motor interference effect from dangerous objects is reflected by the amplitudes of the late component (i.e., P3 amplitudes) rather than by those of the early (P1 and N1) and middle (P2 and N2) latency components. The independent variables were valence (negative, neutral and positive), arousal (medium versus high), and Go/NoGo factor (Go versus NoGo). Kolmogorov–Smirnov tests of normality were also performed on P3 amplitudes in each condition. The results indicated that distributions of P3 amplitudes did not deviate from normality. Accordingly, a three-way repeated-measures ANOVA was used to analyze the effects of the independent variables. The degrees of freedom of the F-ratio were corrected using the Greenhouse-Geisser method, and multiple comparisons were adjusted by the Bonferroni method in the analyses. The effect sizes are presented as partial eta-squared values (η2p) for the ANOVA and as Cohen’s d s for the t-tests.

Figure 1 Topographic plots of the P3 components.

Grand-average topographic plots were calculated based on the mean amplitude in the 300- to 500-ms time window for the Go trials and in the 400- to 600-ms time window for the NoGo trials as a function of valence (negative, positive and neutral) and arousal (medium and high).

Results

Behavioral results

The analysis of RTs revealed significant main effects of valence [F(2, 146) = 50.50, p < 0.001, η2p = 0.41] and arousal [F(1, 73) = 22.01, p < 0.001, η2p = 0.23], together with a significant two-way interaction between valence and arousal [F(2, 146) = 39.05, p < 0.001, η2p = 0.35] (Fig. 2). Subsequent one-way ANOVAs indicated significant main effects of valence at both high [F(2, 146) = 77.59, p < 0.001, η2p = 0.52] and medium [F(2, 146) = 16.91, p < 0.001, η2p = 0.19] arousal levels. Post hoc tests with Bonferroni correction indicated that at the high arousal level, RTs for the negative (492 ± 54 ms; p < 0.001, Cohen’s d = 1.19) and positive valence (492 ± 51 ms; p < 0.001, Cohen’s d = 1.31) conditions were slower than those for the neutral valence condition (474 ± 53 ms), and the difference between the negative and positive valence conditions did not reach significance (p = 1.00, Cohen’s d = 0.01). In contrast, at the medium arousal level, responses for the negative valence condition (496 ± 52 ms) were slower than those for the positive (486 ± 53 ms; p < 0.001, Cohen’s d = 0.64) and neutral valence (488 ± 50 ms; p < 0.001, Cohen’s d = 0.47) conditions, and the difference between the positive and neutral valence conditions did not reach significance (p = 0.76, Cohen’s d = 0.13).

Figure 2 Results of the reaction times.

The figure presents the mean reaction times as a function of valence (negative, positive and neutral) and arousal (medium and high) in Go trials.

Regarding error rates (Fig. 3), the analysis of log-transformed correct rates revealed significant main effects of valence [F(2, 146) = 4.93, p = 0.01, η2p = 0.06] and arousal [F(1, 73) = 5.04, p = 0.03, η2p = 0.07]. Furthermore, a significant two-way interaction between valence and arousal [F(2, 146) = 3.65, p = 0.03, η2p = 0.05] together with a significant three-way interaction among valence, arousal and Go/NoGo factor [F(2, 146) = 5.03, p = 0.009, η2p = 0.06] was identified. Analysis of the simple effect of the significant three-way interaction revealed a significant two-way interaction between valence and arousal in Go trials [F(2, 146) = 7.00, p = 0.02, η2p = 0.09]. Subsequent one-way ANOVAs indicated a non-significant main effect of valence at the high arousal level [F(2, 146) = 0.39, p = 0.65, η2p = 0.005]. However, the main effect of valence reached significance at the medium arousal level [F(2, 146) = 7.54, p = 0.001, η2p = 0.09]. The post hoc analysis indicated that response errors for the negative valence condition (1.72 ± 2.40%) were more than those for the positive (1.04 ± 1.49%; p = 0.003, Cohen’s d = 0.41) and neutral valence (1.11 ± 1.85%; p = 0.02, Cohen’s d = 0.34) conditions, and the difference between the positive and neutral valence conditions did not reach significance (p = 1.00, Cohen’s d = 0.05). In contrast, analysis of the simple effect indicated significant main effects of valence [F(2, 146) = 4.57, p = 0.01, η2p = 0.06] and arousal [F(1, 73) = 7.40, p = 0.008, η2 p = 0.09] in NoGo trials. The post hoc analysis of valence indicated that response errors for the positive (1.19 ±1.24%; p = 0.04, Cohen’s d = 0.30) and negative (1.20 ± 1.29%; p = 0.057, Cohen’s d = 0.28) valence conditions were more and nearly more than those for the neutral valence condition (0.82 ± 1.09%), respectively. The post hoc analysis of arousal indicated that response errors for the high arousal condition (0.91 ± 1.03%) were less than those for the medium arousal condition (1.23 ±1.17%).

Figure 3 Results of the error rates.

The figure presents the mean error rates as a function of valence (negative, positive and neutral), arousal (medium and high) and Go/NoGo factor (Go and NoGo).

ERP results

Grand averages of target-locked ERPs are presented in Fig. 4. The three-way repeated-measures ANOVA results of the centroparietal P3 amplitudes revealed non-significant main effects of valence [F(2, 146) = 2.32, p = 0.10, η2p = 0.03], arousal [F(1, 73) = 1.75, p = 0.19, η2p = 0.02] and Go/NoGo factors [F(1, 73) = 3.21, p = 0.08, η2p = 0.04]. The two-way interaction between valence and arousal reached significance [F(2, 146) = 18.55, p < 0.001, η2 p = 0.20]. However, the two-way interaction between valence and Go/NoGo [F(2, 146) = 2.34, p = 0.10, η2p = 0.03] and that between arousal and Go/NoGo [F(1, 73) = 0.75, p = 0.39, η2p = 0.01], together with the three-way interaction among all factors [F(2, 146) = 2.84, p = 0.06, η2p = 0.04], did not reach significance. To assess the valence effect on different arousal levels, we performed two one-way ANOVAs as a function of valence at the high and medium arousal levels. Specifically, at the medium arousal level, the main effect of valence reached significance [F(2, 146) = 4.41, p = 0.01, η2p = 0.06]. Post hoc tests with Bonferroni correction indicated that the P3 amplitudes for the positive valence (2.81 ± 2.71 µV; p = 0.04, Cohen’s d = 0.30) and neutral valence (2.81 ± 3.03 µV; p = 0.03, Cohen’s d = 0.31) conditions were more positive than those for the negative valence condition (2.53 ± 2.87 µV), and the difference between the positive and neutral valence conditions did not reach significance (p = 1.00, Cohen’s d = 0.003). At the high arousal level, the main effect of valence reached significance [F(2, 146) = 18.43, p < 0.001, η2p = 0.20]. Post hoc tests with Bonferroni correction indicated that the P3 amplitude for the negative valence condition (3.11 ± 2.87 µV) was more positive than that for the neutral valence (2.77 ± 2.72 µV; p = 0.004, Cohen’s d = 0.38) and positive valence (2.52 ± 2.77 µV; p < 0.001, Cohen’s d = 0.70) conditions. Moreover, the P3 amplitude for the neutral valence condition was more positive than that for the positive valence condition (p = 0.03, Cohen’s d = 0.32).

Figure 4 Grand-average target-locked ERPs of the P3 components.

Grand-average target-locked ERPs for Go and NoGo trials were separately presented as a function of valence (negative, positive and neutral) and arousal (medium and high) at the centroparietal area (the average of the C5, C3, C1, Cz, C2, C4, C6, CP5, CP3, CP1, CPz, CP2, CP4, CP6, P5, P3, P1, Pz, P2, P4 and P6 electrodes). The rectangles filled with oblique lines indicate the analyzed time windows for the P3 amplitudes.

Discussion

Overview of the study

This study investigated whether different arousal levels (medium versus high) could modulate the motor interference effect from dangerous objects. The design adopted a motor priming paradigm mixed with a Go/NoGo task, which was consistent with Liu et al. (2017), and further manipulated the valence and arousal of target stimuli referred to Zsido et al. (2020). The target stimuli were selected from the IAPS with a yellow capital letter “M” or “W” superimposed on them as a Go or NoGo signal. Participants were instructed to prepare the ipsilateral hand responses that corresponded to the handedness of the prime and to decide to execute the prepared responses or not according to the Go/NoGo signals.

The hypotheses proposed a modulation effect of different arousal levels on valence factor. Specifically, the motor interference effect should emerge at the medium arousal level with the responses in the negative valence condition executing slower than those in the neutral and positive valence conditions. In contrast, at the high arousal level, the strength of the motor interference effect should increase compared with the medium arousal level. The ERP results predicted more positive P3 amplitudes in the negative valence condition than in the neutral and positive valence conditions at the medium arousal level. Moreover, the differences in the P3 amplitudes between the negative and neutral valence conditions and those between the negative and positive valence conditions should be larger at the high arousal level than the medium arousal level.

The behavioral results of RTs revealed a significant motor interference effect with longer RTs for the negative valence condition than for the neutral and positive valence conditions at the medium arousal level. The results supported the hypothesis that the motor interference effect originated from negative valence rather than the emotional attribute (either negative or positive valence) of the target stimuli. Specifically, only negative background stimuli could distract attention from the main Go/NoGo task at the medium arousal level. At the high arousal level, a significant motor interference effect was also observed with longer RTs for the negative valence condition than for the neutral valence condition. Moreover, the effect size (mean RTs of the negative valence condition minus that of the neutral valence condition) was examined by the two-way interaction between valence (negative versus neutral) and arousal (medium versus high) in Go trials. The results revealed a significant two-way interaction [F(1, 73) = 20.50, p < 0.001, η2p = 0.22], which supports the hypothesis that the effect size of the motor interference effect at the high arousal level (19 ms) is significantly larger than that at the medium arousal level (8 ms). The results support the hypothesis that high arousal stimuli distract many attentional resources from processing the background targets, which may enhance threatening details implied in the negative valence targets. Responses are interfered more strongly in the negative valence condition at the high arousal level than the medium arousal level. However, the difference in RTs between the negative and positive valence conditions did not reach significance, which violates the hypothesis and might suggest that both the negative and positive background stimuli distract attention from the Go/NoGo signals at the high arousal level. The violation of the hypothesis was discussed together with the ERP results in the following paragraphs.

The ERP results present a deeper investigation of the motor interference effect. At the high arousal level, more positive P3 amplitudes emerged in the negative valence condition than the neutral valence condition. As the centroparietal P3 amplitude reflects attentional resource allocation (Isreal et al., 1980), the results support the hypothesis that many attentional resources are assigned to the negative valence condition. Because responding to the Go/NoGo signal is a simple task, remnant attentional resources could be allocated to the background stimuli. This would enhance the processing of the threatening details implied in the negative valence targets, enabling the more dangerous targets. As a more negative valence stimulus may elicit a deeper activation of the aversive motivational system (Lang, Bradley & Cuthbert, 1990; Lang, Davis & Öhman, 2000), the strength of the motor interference effect could be increased. However, at the medium arousal level, less positive P3 amplitudes emerged in the negative valence condition than the neutral valence condition. Assuming that consistent attentional resources are consumed by the Go/NoGo signals, fewer attentional resources could be allocated to the negative valence background stimuli at the medium arousal level than those at the high arousal level, enabling the negative valence targets to be less dangerous; thus, the strength of the motor interference effect could be decreased.

The positive valence was included in the experimental design to exclude an alternative explanation of whether the motor interference effect originates from the emotionality of the background stimuli. Similar result patterns were expected between the positive and negative valence conditions according to this hypothesis. Although identical RTs were observed between the negative and positive valence conditions at the high arousal level, the P3 amplitudes significantly differed. The ERP results indicated that fewer attentional resources were assigned to the positive valence stimuli than to the negative valence stimuli at the high arousal level. A possible reason was proposed based on the valence-arousal conflict theory (Feng et al., 2012; Robinson et al., 2004). The theory suggested that negative valence stimuli evoke the aversive motivational system at a high arousal level. Additionally, subjects might have been more sensitive to the negative valence than the positive valence stimuli, which decreased the P3 amplitude of the positive valence condition. Moreover, the positive valence stimuli force subjects to approach the stimuli (Feng et al., 2012; Robinson et al., 2004). Thus, a conflict exists between high arousal elicited aversive motivation and positive valence evoked approach motivation, which might delay the responses to the positive valence stimuli at the high arousal level. The explanation was also supported by the results at the medium arousal level. According to the valence-arousal conflict theory, the appetitive motivational system will gradually replace the aversive motivational system to dominate emotional cognitive processing along with lower arousal levels. Subjects might be more sensitive to the positive valence than the negative valence stimuli, which increased the P3 amplitude of the positive valence condition at the medium arousal level. When the motivational tendency is congruent between appetitive motivation elicited by medium arousal and approach motivation evoked by positive valence, responses to the positive valence stimuli are accelerated and become close to the neutral valence stimuli at a medium arousal level because the conflict diminishes in the positive valence and medium arousal condition.

The valence-arousal conflict theory could also provide an alternative explanation to the delayed responses and the less positive P3 amplitudes in the negative valence condition compared with those in the neutral valence condition at the medium arousal level. As the appetitive motivational system gradually replaces the aversive motivational system along with lower arousal levels, subjects might be less sensitive to the negative valence stimuli than the neutral and positive valence stimuli, which decreased the P3 amplitude of the negative valence condition at the medium arousal level. Moreover, the motivational tendency is incongruent between appetitive motivation elicited by medium arousal and aversive motivation evoked by negative valence, which might delay the responses to the negative valence stimuli at the medium arousal level. The validation of this alternative explanation requires further investigation.

Note that error rates are not discussed because (1) they are insensitive to the variations of valence and arousal levels. Few errors are made in the simple Go/NoGo task (the mean error rates for each condition were less than 1.72%), which may have caused a ceiling effect; and (2) analyses of the error rates showed only small to medium effect sizes, which might signal false-positive results.

Relationship between the current study and the study by Liu et al. (2017)

The current study was conducted to clarify whether different arousal levels (medium versus high) could modulate the motor interference effect. We hypothesized that arousal levels could modulate valence effect on the behavioral and ERP results. The results revealed a larger motor interference effect at the high arousal level than that at the medium arousal level. Moreover, the centroparietal P3 amplitudes were more positive in the negative valence condition than the neutral valence condition at the high arousal level. Nevertheless, the results of the centroparietal P3 amplitudes were reversed at the medium arousal level. The results expand on those obtained by Liu et al. (2017) by demonstrating that the more positive parietal P3 amplitudes in the dangerous condition than those in the safe condition likely emerged because the dangerous targets not only activated an aversive motivational system (elicited by a negative valence level) but also enhanced the strength of the activation by their high arousal attributes. Only in this manner could the parietal P3 amplitudes be more positive in the dangerous condition than the safe condition in the study by Liu et al. (2017).

Analysis of the centroparietal P3 amplitudes did not reveal significant interactions with the Go/NoGo factor. The results suggest that valence and arousal information is processed regardless of whether the prepared responses were executed (Go trials) or not (NoGo trials). The results are inconsistent with those of Liu et al. (2017), who found a significant dangerous effect (i.e., a more positive parietal P3 amplitude in dangerous conditions than in safe conditions) only when the prepared responses were executed (Go trials). In the NoGo trials, the dangerous effect diminished. The reasons for the different results between the two studies might be due to different perceptual attributes of the background stimuli. Specifically, the colorful animal pictures adopted in the current study are perceptually richer than the gray modern pictures adopted by Liu et al. (2017). The former pictures might reflect possible experiences in daily life, which enables subjects to be more sensitive to the valence and arousal information of the background picture in both the Go and NoGo trials.

Difference between the study by Zsido et al. (2020) and the current study

The current study manipulated valence and arousal factors in accordance with the study by Zsido et al. (2020), identical result patterns were observed at the medium arousal level that responses were delayed when facing a negative valence compared with a neutral valence background stimulus. However, the results differed at the high arousal level. The enlarged RT differnece between the negative and neutral valence conditions at the high arousal level, observed in the current study, however diminished in Zsido et al. (2020). The reason for the discrepancy might attribute to the different difficulties of the main task. Specifically, searching for numbers in a matrix required more attentional resources compared with the Go/NoGo task. Increased attentional resources elicited by the high arousal background stimuli could be assigned to the searching numbers task in priority, which improved performance on the main task. In contrast, discriminating the Go/NoGo signals only required a few attentional resources, and the remaining resources could be assigned to the background stimuli, which enhanced the processing of threatening details implied in the negative valence stimuli. Thus, the motor interference effect was enhanced at the high arousal level in the current study.

Limitations

Although the difference in the arousal scores between the medium (mean = 4.67) and high arousal (mean = 5.68) levels reached significance, the difference was relatively small compared with that in Zsido et al. (2020) because the arousal levels were matched among negative, neutral and positive valences in the animal category of the background stimuli. The results revealed a significant modulation effect of the arousal factor, which verified the effectiveness of the manipulation in distinguishing the medium and high arousal levels. However, we believe that the arousal effect could be more salient if the design increased the difference between the medium and high arousal level. This is a limitation of the current study and should be avoided in further research.

Conclusion

In summary, the current study investigated whether the arousal of target stimuli could modulate the motor interference effect. The behavioral results revealed a significant motor interference effect (mean RTs in the negative valence condition minus those in the neutral valence condition) at the medium arousal level, and the effect size increased at the high arousal level. The results support the hypothesis that the motor interference effect could be modulated by different arousal levels of the stimuli with an increment of the motor interference effect along with a higher arousal level. The results indicate that negative valence stimuli may interfere with the prime elicited motor preparation more strongly at the high arousal level than at the medium arousal level. The underlying cognitive processes are reflected by the results of the centroparietal P3 amplitudes, which indicate that the effect size of the motor interference effect increased because subjects are more sensitive to the negative valence stimuli at the high arousal level than at the medium arousal level, and the activation of the aversive motivational system increased with increasing arousal level. Furthermore, the motor interference effect is related to the negative valence rather than emotionality of the target stimuli because different result patterns emerged between the positive and negative valence conditions. We attempt to relate the findings of the current study to safety management; specifically, increasing the arousal of dangerous stimuli could activate the aversive motivational system. Workers may be more sensitive to dangerous elements in machines when these elements are presented at a high arousal level (e.g., painting bold colors on dangerous elements). Thus, dangerous elements inducing a high state of arousal may increase the motor interference effect, which in turn reduces work-related accidents because responses to dangerous elements have been inhibited.

Supplemental Information

Supplemental Information 1 Behavioral data

Click here for additional data file.

Supplemental Information 2 Centroparietal P3 amplitudes

Centroparietal P3 amplitudes as a function of valence (negative, neutral and positive), arousal (medium versus high) and Go/NoGo factor (Go versus NoGo).

Click here for additional data file.

Supplemental Information 3 Time course of the averaged ERPs

Time course of the averaged ERPs as a function of valence (negative, neutral and positive), arousal (medium versus high) and Go/NoGo factor (Go versus NoGo).

Click here for additional data file.

Additional Information and Declarations

Competing Interests

Author Contributions

Human Ethics

Data Availability

The authors declare there are no competing interests.

Gai Cao and Peng Liu conceived and designed the experiments, performed the experiments, analyzed the data, prepared figures and/or tables, authored or reviewed drafts of the paper, and approved the final draft.

The following information was supplied relating to ethical approvals (i.e., approving body and any reference numbers):

The Medical Ethics Committee at Northwest University approved the study.

The following information was supplied regarding data availability:

Raw data, including behavioral data, centroparietal P3 data, and the time course of the ERPs as a function of valence (negative, neutral and positive), arousal (medium versus high) and Go/NoGo factor (Go versus NoGo), are available in the Supplemental Files.

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
