# Peer review of "Arousal modulates the motor interference effect stimulated by pictures of threatening animals"

_PeerJ, doi:10.7717/peerj.10876_

## Round 0.1 · original submission · Major Revisions

Your manuscript has now been seen by 3 reviewers. You will see from their comments below that while they find your work of interest, some major points are raised. We are interested in the possibility of publishing your study, but would like to consider your response to these concerns in the form of a revised manuscript before we make a final decision on publication. We therefore invite you to revise and resubmit your manuscript, taking into account the points raised. Please highlight all changes in the manuscript text file.

Reviewer 1 ·

Basic reporting

I find the concept of this paper clear, the writing is straightforward and the authors use professional English language. The article structure is good, figures and tables are informative, raw data files are well prepared. The paper is self-contained with relevant results to hypotheses.
My only concern here is that I did not found a sufficient background/context provided, that is the authors did not consider several papers that would be relevant to mention here as these studies also deal with the effect of emotional arousal regarding threatening objects. Especially since in the latter two of these papers, the arousal – motor interference effect is also relevant. Furthermore, since these studies showed very similar results regarding threatening stimuli at low vs high arousal levels. These should also be included in the discussion and the authors could reflect to their results in light of the arousal stimulation effect.

Zsido, A. N., Deak, A., & Bernath, L. (2019). Is a snake scarier than a gun? The ontogenetic–phylogenetic dispute from a new perspective: The role of arousal. Emotion, 19(4), 726.

Zsido, A. N., Bernath, L., Labadi, B., & Deak, A. (2020). Count on arousal: introducing a new method for investigating the effects of emotional valence and arousal on visual search performance. Psychological research, 84(1), 1-14.

Zsido, A. N., Matuz, A., Inhof, O., Darnai, G., Budai, T., Bandi, S., & Csatho, A. (2019). Disentangling the facilitating and hindering effects of threat‐related stimuli–A visual search study. British Journal of Psychology.

Experimental design

The paper introduces a cleverly designed experimental setting that clearly measures what the authors intended to. The research questions are well-defined, relevant and meaningful. The investigation performed is rigorous, the sample size and trial repetition is great, and the authors did a really great job checking the emotional variables of the stimuli used. The research presented here definitely fills a gap. Methods are described with sufficient details and information so that further studies can replicate it.

Validity of the findings

Please report the full statistical results for the post-hoc tests (not just p values).
Some of the analyses (e.g. error rates) showed only small to medium effect sizes. This, with the high sample size and p values close to .05 could signal false positive results. This should be explicitly mentioned in the discussion.
The studies I referred to earlier should be mentioned in the discussion since they showed very similar results.
I really liked the occupational-applied framework of this research paper and I think that is a very important thing to put basic research in such a perspective. Yet, as PeerJ reviewer suggestion highlight: Speculation is welcome, but should be identified as such. So please make sure this is true for the discussion of the results.

Additional comments

No further comment.

Reviewer 2 ·

Basic reporting

The manuscript is well written and clear in each part (intro, method, results, and discussion). The literature cited in the introduction is appropriate, but in some cases, unfortunately, misinterpreted and misguiding (see more comments on this in the next sections). The figures are relevant to support authors claims, even though the supplemental material does not appear to be complete or easy to access for everyone. It would have been useful, for example, to attach a translated version on the consent form or the approval documentation. As for the ERPs, I appreciate the inclusion of the average amplitude for each subject, but a text file containing the time course of the ERPs would have been probably more useful because it allows to replicate the figures reported in the manuscript and the average of a certain time window on interest can be easily calculated from there.

Experimental design

The experimental design is conceptually clever, and the details well described. However, the orthogonal manipulation of valence and arousal is not supported by the literature and there are several evidences in this regard:
1. The authors are correctly referring many times to Lang’s studies, but in that model arousal and valence are NOT orthogonal. As the authors described in the introduction, valence indicates the motivational system that is supposed to be activated in a given situation (appetitive or defensive), and arousal the level of activation. Arousal and valence are expected to covary: as valence ratings increase, arousal ratings increase as well. This is also the reason of the so called ‘boomerang shape’ for the affective space defined by the IAPS.
And in the affective space defined by the IAPS, there are no neutral stimuli high in arousal. It simply cannot happen. In fact, the authors basically selected stimuli that would be considered NEUTRAL in almost every study. In a scale that varies from 1 to 9 for arousal, the authors selected stimuli with average scores of 4.5 or 5.5 (values reported in Table 2). The fact that there is a significant difference between these values it does not mean that the manipulation was successful. High arousing stimuli are considered pictures with scores of 6.5-7 and above, and low arousing stimuli are considered pictures with a score of ~2.
2. There is no proof of positivity offset (advantage of positive stimuli at low arousal) and negativity bias (advantage of negative stimuli over positive stimuli at high arousal) in ERPs literature. This is only a conceptual model developed by Cacioppo in the ’90 that has been largely dis-proofed (see Weinberg & Hajcak, 2010; Beyond Good and Evil: The Time-Course of Neural Activity Elicited by Specific Picture Content). Over 20 years of ERPs research found that, when stimuli are balanced for arousal, there is no negative bias in P3-LPP component.
3. The authors misinterpreted, in part, the concepts of positivity offset and negativity bias. These concepts does not necessarily decries differences between positivity and negativity in terms of absolute amplitude. Instead, Cacioppo (Cacioppo, Gardener, Berntson, 1997; Beyond Bipolar Conceptualizations and Measures: The Case of Attitudes and Evaluative Space) conceptualized the negativity bias, from Miller studies on approach avoidance gradients from the ’50, as a difference in the transfer functions of positivity and negativity (from the article cited above, at page 13): “ With each unit of activation, the change in negative motivational output is larger than the change in positive motivational output”. In short, the difference between positivity and negativity is essentially in the slope, when activation (arousal) varies from low to high).

Validity of the findings

All the methodological procedures are perfect and well described. From a methodological point of view, the results are perfect. However, the validity of the findings is unfortunately questioned by the conceptual limitations. The orthogonal manipulation of valence and arousal would not be positively accepted in the field, together with the problematic conceptualization of the negativity bias. An additional issue is that the authors found their effects in Fz and Cz, while all the ERPs studies cited in the introduction in reference to picture viewing (e.g., Cuthbert et al., 2000; Keil et al., 2002; Polich, 2007; Sabatinelli et al., 2007; Schupp et al., 2003) are in Pz or cluster of sensors around Pz. This is another point that would probably not be well received by experts in the field.

Additional comments

Despite being a well-written article with a good methodological approach for ERP analysis, there are major conceptual issues that does not make the study suitable for publication and cannot be addressed in a revision. I would invite the authors to re-consider the orthogonal manipulation of valence and arousal and eventually replicate the same study by keeping in mind how the negativity bias was originally conceptualized.

Reviewer 3 ·

Basic reporting

no comment

Experimental design

no comment

Validity of the findings

no comment

Additional comments

The present study utilized the P3 component, reaction time and error rate to examine whether arousal and/or valence drives the allocation of attentional resources. The authors found that the parietal P3 during the negative valence condition was increased in the high arousal condition compared to the low arousal condition. In the positive valence condition, the parietal P3 was reduced in the high arousal condition compared to the low arousal condition. Similar results were obtained at the frontal and central electrode sites. The question the study tries to answer seems important but the paper needs some clarity to have a bigger impact. Overall, the paper is difficult to follow.

1) The authors provide a lot of detail in the introduction which is good and I appreciate the authors grounding their hypotheses in theory. In doing so, the hypotheses are numerous and difficult to obtain. I think the introduction would be clearer if it was re-ordered and the hypotheses were made more obvious. Also, the authors should make it more obvious why this study is needed. This information was hard to obtain from a first reading of the introduction. I suspect readers would have similar difficulties.

2) The authors mentioned that large artifacts were manually deleted. What was entailed in this process? Who was in charge of cleaning the data?

3) There a mistake in line 415. It states positive > neutral > neutral.

4) In the discussion, line 508, the authors say "In the emotional field..". Please clarify that.

5) Please tie in the current study more explicitly with the previous study (Liu et al 2017) in the discussion. The discussion could also use some reorganizing.

6) I don't believe the authors are testing the hypothesis that valence has a bigger effect on frontal/central sites than parietal sites. I don't see any comparisons of the P3 within the valence conditions are different electrodes. I could be wrong. The information was hard to obtain.

7) The abstract doesn't mention anything about the p3 finding, except the finding in the Liu paper. Authors should mention it in the abstract. Or alternatively, it might make sense to only look at the P3 at Pz since that is what the Liu et al study seemed to have done.

---

## Round 0.2 · accepted · Accept

Thank you for the revised manuscript and response letter. I am pleased to inform you that your manuscript has been accepted for publication. However, before publication, please take into account the suggestions and corrections provided by Reviewer 3.

Reviewer 1 ·

Basic reporting

Language is clear and professional, sufficient background provided, the structure is good.

Experimental design

The experimental design is sound, hypotheses/research questions are well defined, details are sufficient.

Validity of the findings

The findings are novel and interesting, the data support the claims the authors make, the conclusions are well stated. I think that the results will have a great impact on the field and attract many citations in the future.

Additional comments

The authors did a really great job responding to my comments. I wholeheartedly recommend this manuscript to be accepted without further modifications. I am looking forward to reading more papers form the authors investigating this topic.

Reviewer 3 ·

Basic reporting

No comment

Experimental design

No comment

Validity of the findings

No comment

Additional comments

I would like to thank the authors for their work on changing the manuscript according to my comments. I appreciate their efforts. I have just two small comments below.


1) Please replace insignificant with non-significant in the method and results section.

2) The pooling of 21 channels to create a centroparietal P3 seems excessive. The authors point to figure 1 as their justification; however, the activity is maximal in a smaller number of channels than the authors used. I guess I would suggest that the authors check the prior literature to see what makes sense to use for their pooling average. Looking at figure 1, I'd only pool at the sites very near Pz (including Pz).